# RATE-DISTORTION OPTIMIZED POST-TRAINING QUANTIZATION FOR LEARNED IMAGE COMPRESSION

## ABSTRACT

Quantizing floating-point neural network to its fixed-point representation is crucial for Learned Image Compression (LIC) because it ensures the decoding consistency for interoperability and reduces space-time complexity for implementation. Existing solutions often have to retrain the network for model quantization which is time consuming and impractical. This work suggests the use of Post-Training Quantization (PTQ) to directly process pretrained, off-the-shelf LIC models. We theoretically prove that minimizing the mean squared error (MSE) in PTQ is suboptimal for compression task and thus develop a novel Rate-Distortion (R-D) Optimized PTQ (RDO-PTQ) to best retain the compression performance. Such RDO-PTQ just needs to compress few images (e.g., 10) to optimize the transformation of weight, bias, and activation of underlying LIC model from its native 32-bit floating-point (FP32) format to 8-bit fixed-point (INT8) precision for fixed-point inference onwards. Experiments reveal outstanding efficiency of the proposed method on different LICs, showing the closest coding performance to their floating-point counterparts. And, our method is a lightweight and plug-and-play approach without any need of model retraining which is attractive to practitioners.

## 1 INTRODUCTION

Compressed images are used vastly in networked applications for efficient information sharing, which continuously drives the pursuit of better compression technologies for the past decades (Wallace, 1992; Sullivan et al., 2012; Bross et al., 2021). Built upon the advances of deep neural networks (DNNs), recent years have witnessed the explosive growth of image compression solutions (Ballé et al., 2018; Minnen et al., 2018; Chen et al., 2021; Cheng et al., 2020; Hu et al., 2021; Lu et al., 2022) with superior efficiency to well-known rules-based JPEG (Wallace, 1992), HEVC Intra (BPG) (Sullivan et al., 2012), and even Versatile Video Coding Based Intra Profile (VVC Intra) (Bross et al., 2021).

Nevertheless, existing learned image compression (LIC) approaches typically adopt the floating-point format for data representation (e.g., weight, bias, activation), which not only consumes excessive amount of space-time complexity but also brings up the platform inconsistency and decoding failures (He et al., 2022). To tackle these for practical application, model quantization is usually applied to generate fixed-point (or integer) LICs (Ballé et al., 2018; Hong et al., 2020; Sun et al., 2021).

Popular Quantization-Aware Training (QAT) (Bhalgat et al., 2020; Le et al., 2022; Sun et al., 2021) was mainly used in (Ballé et al., 2018; Hong et al., 2020; Sun et al., 2020; 2021) to transform floating-point LIC to its fixed-point representation. Such methods requires model re-training with the full access of labels which is expensive and impractical.

Recently, Post-Training Quantization (PTQ) (Nagel et al., 2020; 2021) offered a lightweight and plug-and-play solution to directly quantize pretrained, off-the-shelf network models without requiring model retraining. However, such PTQ scheme was mostly dedicated for high-level vision tasks as studied in (Choukroun et al., 2019; Liu et al., 2021). This work therefore extends the use of PTQ to image compression model quantization.

We theoretically prove that only minimizing the quantization error (e.g., MSE) in PTQ for LIC model quantization as other vision tasks (Choukroun et al., 2019; Liu et al., 2021) is sub-optimal from the compression perspective because of localized non-monotonic relation between the quantization error and rate-distortion performance of the image compression task. We thus propose the Rate-Distortion Optimized PTQ (RDO-PTQ) for LIC model quantization.

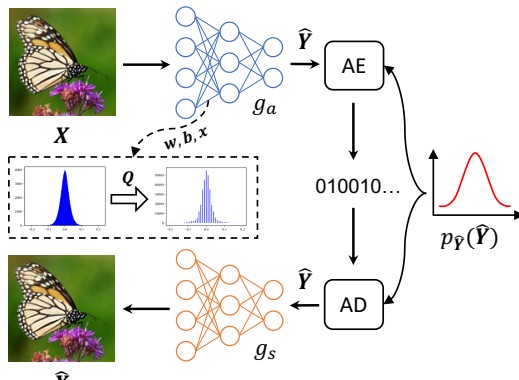

Considering the optimization complexity, the RDO-PTQ is executed from one network layer[1] to anther (e.g., layerwisely) to process weight, bias and activation in either convolutional or self-attention computation to respectively determine their proper ranges for quantization. In current implementation, it just compresses a tiny calibration image set (e.g., less than 10 images) to optimize relevant quantization factors like range, offset, etc for a fixed-point model.

Figure 1: *Learned Image Compression (LIC)*. $g_a$ ($g_s$) is main encoder (decoder); AE/AD is arithmetic encoding/decoding using $p_{\hat{Y}}(\hat{Y})$. Either convolution or self-attention is used to derive $\hat{Y}$ of input $X$. Model quantization $Q$ is applied at every layer (convolutional or self-attention) to transform weight $w$, bias $b$ and activation $x$ in native FP32 to INT8 precision.

Given that the distribution of both weight and activation varies across channels at each network layer, the range is adapted channel-wisely besides the layer-wise adaptation (see Fig. 3); Besides the range determination of the bias, a bias rescaling is applied to ensure the computation using INT8 data tensor strictly.

**Contribution.** 1) We suggest the use of PTQ to quantize LIC model for a lightweight, plug-and-play solution by just compressing fewer image samples to derive the fixed-point model without any model retraining; 2) Both rate and distortion metrics are optimized jointly at compression task inference stage to determine proper ranges of weight, bias and activation for quantization in proposed RDO-PTQ; 3) Our method is generalized to a variety of LIC models, demonstrating the closest compression efficiency between native FP32 and corresponding quantized INT8 model.

## 2 RELATED WORK

**Learned Image Compression (LIC).** As shown in Fig. 1, popular LICs are mainly built upon the Variational Auto-Encoder (VAE) architecture to find rate-distortion optimized compact representation of input image. In Ballé et al. (2018), on top of the GDN (Generalized Divisive Normalization) based nonlinear transform, a hyper prior modeled by a factorized distribution was introduced to better capture the distribution of latent features. Shortly, the use of joint hyper prior and autoregressive neighbors for entropy context modeling was developed in Minnen et al. (2018), demonstrating better efficiency than the BPG (e.g., a HEVC Intra implementation).

Later then, stacked convolution with simple ReLU was used in (Cheng et al., 2020; Chen et al., 2021) to replace GDN and the attention mechanism was augmented for better information embedding, which, for the first time, outperformed the VVC Intra. Recalling that the principle behind the image coding is to find content-dependent models (e.g., transform, statistical distribution) for more compact representation, apparently, solutions simply stacking convolutions are not capable of efficiently characterizing the content dynamics because of the fixed receptive field and fixed network parameters of a trained convolutional neural network (CNN). To enable the content-adaptive dynamic embedding, self-attention mechanism was extended in (Qian et al., 2021; Lu et al., 2022; Lu & Ma, 2022). As extensively studied in (Lu et al., 2022; Lu & Ma, 2022), an integrated convolution

---

[1]For simplicity, we refer the "network layer" to as the "layer".

and self-attention could not only provide performance improvements to the VVC intra on various datasets, but also reduces the space-time complexity significantly.

As shown subsequently, three popular open-resource LICs are used to exemplify the efficiency of our RDO-PTQ, including the Minnen2018 (Minnen et al., 2018), Cheng2020 (Cheng et al., 2020) and Lu2022 (Lu et al., 2022).

**LIC Model Quantization.** Although learning-based solutions remarkably improved the performance for various tasks, their native floating-point representation incurred a serial problems for practical enabling, including the excessive amount of space-time complexity consumption, nondeterministic inconsistency across heterogeneous platforms, etc. These issues are more severe for image compression task because the guarantee of interoperability across a variety of devices is the vital functionality of a LIC solution. As reported, a small numerical rounding-off error in floating-point computation may lead to decoding failures or incorrect reconstructions (Ballé et al., 2018).

A pioneer exploration was made in Ballé et al. (2018) to train an integer LIC to resolve platform inconsistency and decoding failures. Subsequently, the quantization of convolutional weights was specifically treated with in Sun et al. (2020; 2021). Almost at the same time, Hong et al. (Hong et al., 2020) suggested layer-wise range-adaptive quantization (RAQ) for both weights and bias, and linear scaling for feature activation. Recently, Le et al. (2022) quantized the transformer-based learned codec and realized the real-time decoding on a mobile device. Note that these methods require the full access of labels for model retraining when performing the quantization, which is impractical and expensive to some extent (Nagel et al., 2021).

Recently, PTQ attracts intensive attentions (Nagel et al., 2021) because it is a push-button approach without model retraining, which is presumably applicable to any off-the-shelf, pretrained neural networks. Unfortunately, a majority of PTQ studies were still devoted to high-level vision tasks as in (Choukroun et al., 2019; Nagel et al., 2020; Liu et al., 2021), where, more importantly, they mainly minimized quantization induced MSE (Mean Square Error). Given the advantages of PTQ, this work extends it to the image compression task; Since image compression pursues the optimal rate-distortion performance, a novel rate-distortion optimized PTQ is developed to fulfill the purpose.

## 3 QUANTIZATION FUNDAMENTAL

For an arbitrary floating-point value $x_{float}$ (e.g., weight, bias or activation of either a convolutional or self-attention layer), it is quantized using:

$$x_{int} = clip(\lfloor \frac{x_{float}}{s_x} \rceil + z_x, -2^{b-1} + 1, 2^{b-1} - 1), \tag{1}$$

where $\lfloor \cdot \rceil$ rounds the input to its nearest integer. $z_x$ is the offset and $s_x$ is the linear scaling factor.

Each $x_{int}$ is mapped to a fixed-point $\hat{x}$ for inference (Hong et al., 2020) or simulating the effect of integer quantization to avoid gradient vanishing (Dai et al., 2021) in training. Thus, the mapping is often formulated as

$$\hat{x} = s_x \cdot x_{int} - z_x. \tag{2}$$

Such mapping is also used in our RDO-PTQ for optimizing quantization factors by examining the compression performance of a tiny set of images in task inference stage to derive the corresponding fixed-point model.

As pointed out in Nagel et al. (2021), $z_x$ typically makes integer computation more complicated. Therefore, practical accelerators usually apply the symmetric quantization assuming $z_x = 0$. In this way, we only need to define the scale factor $s_x$ and bit width precision $b^s$. In general, $b^s$ can be predefined, and $s_x$ is deduced using

$$s_x = \Gamma(\frac{r_x}{2^{b-1} - 1}; n_r, b^s), \tag{3}$$

where $r_x$ represent the dynamic range of $x$, $n_r$ is reserved number of decimal digits and $\Gamma(\cdot)$ is defined below

$$\Gamma(s; n_r, b^s) = clip(s \cdot 2^{n_r}, -2^{b^s-1}, 2^{b^s-1} - 1) \times 2^{-n_r}. \tag{4}$$

In practice, these functions can be implemented using bit shift operation.

## 4  RATE-DISTORTION OPTIMIZED PTQ

### 4.1  FROM QUANTIZATION ERROR TO R-D LOSS

To derive proper $s_x$ for model quantization as in Eq. (3), existing works mostly minimize the square error between vectorized elements $x$ in FP32 and quantized $\hat{x}$ in INT8, e.g.,

$$s_x = \arg\min_{s_x} \|x - \hat{x}\|^2. \tag{5}$$

The use of Eq. (5) generally assumes the monotonic relation between quantization error induced distortion $D$ (e.g., MSE) and compression efficiency measured by the R-D metric $J = R + \lambda D$ (Davisson, 1972). Apparently, such assumption does not hold because of inevitable rate contribution to $J$.

**Theoretical Justification.** Assuming a trained compression task model $J(x, w)$ with floating-point activation $x$ and weight $w$, its fixed-point model used for inference can be formulated as $J(x + \Delta x, w + \Delta w)$. $\Delta w$ and $\Delta x$ are quantization noises for $w$ and $x$ respectively. To simplify the deduction, the bias term is neglected since it can be absorbed into $w$ by appending a unit term to $x$ as studied in Botev et al. (2017).

Then the model quantization induced performance loss of underlying task model is:

$$\Delta J = E\left[J(x + \Delta x, w + \Delta w) - J(x, w)\right]$$
$$\approx E\left[\left(\left[\begin{array}{cc} \frac{\partial J}{\partial x} & \frac{\partial J}{\partial w} \end{array}\right] + \frac{1}{2}[\Delta x \ \Delta w]\, H_{x,w}\right)\left[\begin{array}{c} \Delta x \\ \Delta w \end{array}\right]\right], \tag{6}$$

where $H_{x,w}$ is the Hessian matrix, and high-order terms are ignored for simple derivation. For a converged model, its gradient is close to 0, e.g., $[\frac{\partial J}{\partial x} \ \frac{\partial J}{\partial w}] = 0$, yielding

$$\Delta J \approx \frac{1}{2}E\left[\left[\begin{array}{c} \Delta x \\ \Delta w \end{array}\right]^T \cdot H_{x,w} \cdot \left[\begin{array}{c} \Delta x \\ \Delta w \end{array}\right]\right]. \tag{7}$$

Subsequently, Eq. (7) can be further expanded as:

$$\Delta J \approx \frac{1}{2}E[\frac{\partial^2 J}{\partial x^2}\Delta x^2 + \frac{2\partial^2 J}{\partial x \partial w}\Delta x \Delta w + \frac{\partial^2 J}{\partial w^2}\Delta w^2]. \tag{8}$$

As seen, the R-D loss not only depends on the quantization error, e.g., $\Delta x^2$ and $\Delta w^2$, but also is related to the second-order derivatives of $J$. Particularly, even having all positive second-order derivatives, the different sign of $\Delta x$ and $\Delta w$ would lead to a negative cross term $\Delta w \Delta x$, suggesting that a larger absolute error of quantization may not lead to a larger R-D loss. Similar observation was also discussed in Nagel et al. (2020).

As a toy example, suppose a symmetric Hessian matrix as

$$H_{x,w} = \left[\begin{array}{cc} 1 & 0.5 \\ 0.5 & 1 \end{array}\right]. \tag{9}$$

Then, the R-D loss can be rewritten as

$$\Delta J \approx \frac{1}{2}E\left[\Delta x^2 + \Delta x \Delta w + \Delta w^2\right]. \tag{10}$$

If we happen to have a case with a larger quantization error, e.g., $[\Delta x, \Delta w] = [0.4, -0.4]$, and the other scenario with a smaller quantization error $[0.3, 0.3]$, the R-D loss is $0.08$ and $0.135$ respectively. As seen, larger quantization error gives smaller R-D loss.

**Localized Non-monotonic Behavior.** In the meantime, we visualize the absolute quantization error of weight, e.g., $\Delta w$ with compression task loss measured by the $\Delta J$, which further confirms the theoretical justification aforementioned. As seen in Fig. 2, although over a wide range of $\Delta w$, it is monotonically related with the R-D loss $\Delta J$ in compression task; it presents localized non-monotonic behavior where the minimization of quantization error does not lead to the minimum of $\Delta J$, which, consequently, degrades the overall compression performance of underlying LIC model if we pursue the minimization of quantization error for model quantization.

As a result, to best retain the identical compression efficiency of quantized LIC as its floating-point counterpart, we propose to optimize the PTQ from the rate-distortion perspective for underlying compression task.

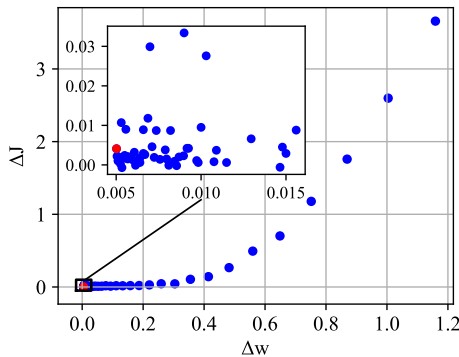

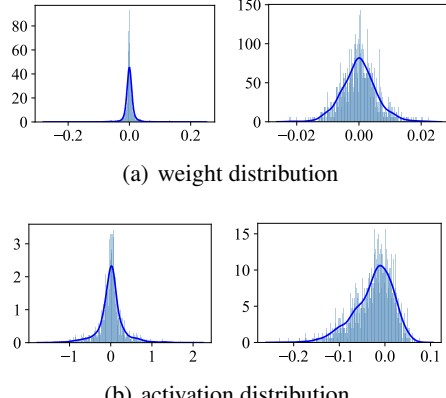

(a) weight distribution

(b) activation distribution

Figure 2: *Compress Task Loss vs. Quantization Error.* Localized non-monotonic behavior is presented where the minimization of quantization error $\Delta \boldsymbol{w}$ does not lead to the minimal loss of rate-distortion metric ($\Delta J$).

Figure 3: Exemplified weight and activation distribution of two channels for a given layer. Other layers exhibit similar distribution from a channel to another.

## 4.2 RATE-DISTORTION OPTIMIZED SCALING

As for the compression task, the optimization target in PTQ is to minimize the R-D loss by considering the distortion $D$ and bite rate $R$ jointly (Davisson, 1972). For a typical VAE structure with conditional context modeling, its R-D loss is

$$J = \lambda \cdot D + R_{\hat{\boldsymbol{Y}}} + R_{\hat{\boldsymbol{Z}}} = \lambda \cdot D(\boldsymbol{X}, \hat{\boldsymbol{X}}) + E[-\log_2(p_{\hat{\boldsymbol{Y}}}(\hat{\boldsymbol{Y}}|\hat{\boldsymbol{Z}}))] + E[-\log_2(p_{\hat{\boldsymbol{Z}}}(\hat{\boldsymbol{Z}}))], \quad (11)$$

where $R = R_{\hat{\boldsymbol{Y}}} + R_{\hat{\boldsymbol{Z}}}$ is the total bit rate consumed by the latent feature $\hat{\boldsymbol{Y}}$ and hyper prior $\hat{\boldsymbol{Z}}$, and $D$ is the distortion between the original $\boldsymbol{X}$ and its reconstruction $\hat{\boldsymbol{X}}$, which is typically computed using MSE or MS-SSIM. Besides, $p_{\hat{\boldsymbol{Y}}}$ and $p_{\hat{\boldsymbol{Z}}}$ are the probability distribution of respective $\hat{\boldsymbol{Y}}$ and $\hat{\boldsymbol{Z}}$ for entropy coding. We adapt Lagrange multiplier $\lambda$ for various bit rates as they do in (Minnen et al., 2018; Cheng et al., 2020; Lu et al., 2022).

For any pretrained floating-point LIC model $J_0$, the examination of the proposed RDO-PTQ is defined below to derive proper scaling factor for the quantization of activation, weight and bias, e.g.,

$$s_{\boldsymbol{x}}, s_{\boldsymbol{w}}, s_{\boldsymbol{b}} = \underset{s_{\boldsymbol{x}}, s_{\boldsymbol{w}}, s_{\boldsymbol{b}}}{\arg\min} \|\widehat{J} - J_0\|^2 = \underset{s_{\boldsymbol{x}}, s_{\boldsymbol{w}}, s_{\boldsymbol{b}}}{\arg\min} \|(\widehat{R} + \lambda \cdot \widehat{D}) - (R_0 + \lambda \cdot D_0)\|^2. \quad (12)$$

It is impractical to optimize all layers of a LIC model at once because the accumulation of quantization errors from layer to layer (e.g., forward and backprop) makes it extremely difficult for such a great amount of parameters (e.g., weight, bias, activation) to converge fastly and reliably. We therefore suggest to apply the layer-wise quantization to progressively optimize the parameters. Usually, parameters at former layer are fixed for the optimization of successive layers. As for the parameter quantization at $l$-th layer, all parameters from the very first to $(l-1)$-th layer are already quantized and fixed for optimization; while floating-point parameters from $(l+1)$-th layer to the last layer are kept without change. Thus, Eq. (12) is reformulated as:

$$s_{\boldsymbol{x}}^l, s_{\boldsymbol{w}}^l, s_{\boldsymbol{b}}^l = \underset{s_{\boldsymbol{x}}^l, s_{\boldsymbol{w}}^l, s_{\boldsymbol{b}}^l}{\arg\min} \|\widehat{J}(\hat{\boldsymbol{x}}^l, \hat{\boldsymbol{w}}^l, \hat{\boldsymbol{b}}^l) - J_0(\boldsymbol{x}^l, \boldsymbol{w}^l, \boldsymbol{b}^l)\|^2, \quad (13)$$

where $\hat{\boldsymbol{x}}^l = \Lambda(\hat{\boldsymbol{w}}^{l-1}\hat{\boldsymbol{x}}^{l-1} + \hat{\boldsymbol{b}}^{l-1})$ with $\Lambda(\cdot)$ as a nonlinear activation function like ReLU. Besides, inspired by AdaQuant (Hubara et al., 2020) , AdaRound (Nagel et al., 2020) and BRECQ (Li et al., 2020), a continuous variable $\boldsymbol{v}^l$ is added to the weight parameter for better rounding,

$$\hat{\boldsymbol{w}}^l = Q(\boldsymbol{w}^l + \boldsymbol{v}^l), \quad (14)$$

where $Q(\cdot)$ is the quantization function. Note that the same operation for the weight is applied to the bias as well. Further details are provided in the supplementary material.

### 4.3 DYNAMIC RANGE DETERMINATION

In Eq. (3), when the bit width precision is given, scaling factor, e.g., $s_{\boldsymbol{w}}^l$, $s_{\boldsymbol{b}}^l$ or $s_{\boldsymbol{x}}^l$, is only related to the dynamic range $r^l$ of corresponding data tensor, e.g., $\boldsymbol{w}^l$, $\boldsymbol{b}^l$, or $\boldsymbol{x}^l$. For the uniform quantization used mostly, the derivation of scaling factor is equivalent to the determination of the dynamic range. In general, weight, bias and activation assume the similar Gaussian distribution (Hong et al., 2020), and their ranges are approximated below.

**Weight.** As shown in Fig. 3, weight distribution not only varies from one layer to another of a given LIC model, but also from one channel to another at a given layer. This suggests us to model the range of weight tensor using

$$r_{\boldsymbol{w}}^{l,k} = N_{\boldsymbol{w}}^{l,k} \cdot max(|\boldsymbol{w}^{l,k}|), \qquad (15)$$

with $max(|\boldsymbol{w}^{l,k}|)$ directly computed from $k$-th channel weights at $l$-th layer. As seen, our method mainly attempts to determine the proper $N_{\boldsymbol{w}}^{l,k}$ channel-wisely for weight quantization. Although Sun et al. (Sun et al., 2021) did the similar channel-wise grouping to clip weights for fixed-point processing, model retraining was required for finetuning, which largely differs from our PTQ solution.

**Activation.** Feature activations are closely related to the original image input. However, Hong et al. (Hong et al., 2020) enforced a fixed dynamic range per layer, e.g., $[\mu - 3\sigma, \mu + 3\sigma]$ to normalize the activation, which is apparently sub-optimal without thoroughly considering the dynamics of input content. After carefully examining the activation distribution, the channel-wise variations in activation tensor are also considered, leading to

$$r_{\boldsymbol{x}}^{l,k} = N_{\boldsymbol{x}}^{l,k} \cdot max(|\boldsymbol{x}^{l,k}|). \qquad (16)$$

**Bias.** First, the range of bias is approximated layerwisely using

$$r_{\boldsymbol{b}}^l = N_{\boldsymbol{b}}^l \cdot max(|\boldsymbol{b}^l|), \qquad (17)$$

to determine the scaling factor $s_{\boldsymbol{b}}^l$ using Eq. (3).

Typically, bias term is augmented with the product of weight and activation to output feature activation of current $l$-th layer as the input of next $(l+1)$-th layer. Even though we enforce INT8 precision for all data tensor, in practice, a 32-bit accumulator is often used to host intermediate data $\hat{\boldsymbol{x}}_{imd}^l$ to avoid potentially data overflow. This suggests:

$$\hat{\boldsymbol{x}}_{imd}^l = \hat{\boldsymbol{w}}^l \cdot \hat{\boldsymbol{x}}^l + \hat{\boldsymbol{b}}^l. \qquad (18)$$

Here, we wish to simply scale 32-bit $\hat{\boldsymbol{x}}_{imd}^l$ to have 8-bit $\hat{\boldsymbol{x}}^{l+1}$ to input the $(l+1)$-th layer.

By combining Eq. (2), as long as we have

$$\tilde{\boldsymbol{b}}_{int}^l = \lfloor \frac{s_{\boldsymbol{b}}^l}{s_{\boldsymbol{w}}^l \cdot s_{\boldsymbol{x}}^l} \cdot \boldsymbol{b}_{int}^l \rceil. \qquad (19)$$

we then arrive at

$$\hat{\boldsymbol{x}}_{imd}^l = s_{\boldsymbol{w}}^l \cdot \boldsymbol{w}_{int}^l \cdot s_{\boldsymbol{x}}^l \cdot \boldsymbol{x}_{int}^l + s_{\boldsymbol{b}}^l \cdot \boldsymbol{b}_{int}^l = s_{\boldsymbol{w}}^l \cdot s_{\boldsymbol{x}}^l \cdot (\boldsymbol{w}_{int}^l \cdot \boldsymbol{x}_{int}^l + \tilde{\boldsymbol{b}}_{int}^l), \qquad (20)$$

which shows that we can simply scale 32-bit $\hat{\boldsymbol{x}}_{imd}^l$ to derive 8-bit input $\boldsymbol{x}_{int}^{l+1}$, i.e., $\boldsymbol{x}_{int}^{l+1} = \frac{\hat{\boldsymbol{x}}_{imd}^l}{s_{\boldsymbol{x}}^{l+1}}$.

Often time, $\frac{s_{\boldsymbol{b}}^l}{s_{\boldsymbol{w}}^l \cdot s_{\boldsymbol{x}}^l} \gg 1$ in Eq. (19) which would not affect the task performance due to rounding operation. We call the operation defined in Eq. (19) bias rescaling which is used to enforce strict INT8 computation layer by layer.

On the contrary, many existing works either simply assume the INT32 precision, or brutally enforce zeros, for bias. Such inappropriate processing of bias may lead to catastrophic results. For example, having the bias in INT32 precision may cause data overflow of the INT32 accumulator; while setting bias as zero would degrade the model performance significantly. More details can be found in supplementary materials.

---

**Algorithm 1** RDO-PTQ for Learned Image Compression

---

**Input:** Floating-point model; Calibration image set
**Output:** The quantized model; Optimized scaling factors
 1: **for** $l$ in $\{\mathcal{L}\}_{l=1}^{L}$ **do**
 2:    **repeat**
 3:      **repeat**
 4:        Initial $s_{\boldsymbol{w}}^{l,k}, s_{\boldsymbol{b}}^{l}$
 5:        Quantize l-th layer and forward propagation
 6:        Update $N_{\boldsymbol{w}}^{l,k}, N_{\boldsymbol{b}}^{l}$ by SGD (Hinton et al., 2012; Kingma & Ba, 2014)
 7:      **until** $s_{\boldsymbol{w}}^{l,k}, s_{\boldsymbol{b}}^{l} = \arg\min\|(\hat{R} + \lambda \cdot \hat{D}) - J_0\|^2$
 8:      **repeat**
 9:        Initial $s_{\boldsymbol{x}}^{l,k}$
10:        Forward propagation
11:        Update $N_{\boldsymbol{x}}^{l,k}$ by SGD
12:      **until** $s_{\boldsymbol{x}}^{l,k} = \arg\min\|(\hat{R} + \lambda \cdot \hat{D}) - J_0\|^2$
13:    **until** Convergence or excess limitation
14: **end for**

---

## 4.4 SUMMARY

The overall RDO-PTQ process is summarized in Algorithm 1. We optimize the model from one network layer to another where dynamic range and scale factor for weight and bias are processed first, and then the activation. When the rate-distortion loss reaches at its minimum, the optimization for $l$-th layer is completed and all settings are fixed for the optimization of $(l + 1)$-th layer. Such process continues until the last layer of the model. Note that except for the bias that we apply the layer-wise processing with rescaling, we perform channel-wise quantization for both weight and activation parameters.

## 5 EXPERIMENTS

Extensive experiments are conducted to report the efficiency of the proposed RDO-PTQ for LIC model quantization.

### 5.1 COMPARISON SETUP

**Pretrained FP32 LICs.** We choose three popular LIC models in their native FP32 format for evaluation, namely Minnen2018 (Minnen et al., 2018), Cheng2020 (Cheng et al., 2020) and Lu2022 (Lu et al., 2022). For instance, Minnen2018 is a seminal work first introducing the joint exploration of hyperprior and autoregressive neighbors for entropy coding of latent features; and Cheng2020 is one of several works which first proposed to apply attention mechanism to better aggregate information. Both Minnen2018 and Cheng2020 rely on stacked convolutions while Lu2022 is one of several earlier attempts that uses both convolution and self-attention. Subsequent results report the generalization of our RDO-PTQ to these representative models regardless of its key unit (e.g., convolution or self-attention, ReLU or GDN, etc), which is attractive to practical applications.

Pretrained models are directly obtained from their open resource websites[2][3], respectively. Models used in comparison are trained using MSE loss for distortion measurement in R-D metric. Results for MS-SSIM loss trained models are provided in supplemental material. For each model, six bitrates are experimented by setting six different $\lambda$s, e.g., $\{0.0018, 0.0035, 0.0067, 0.013, 0.025, 0.0483\}$ directly without any finetuning.

**Alternative PTQs.** Given that Lu2022 currently demonstrates the leading compression efficiency, we mainly use it as the baseline to implement other model quantization schemes. Unfortunately, to the best of our knowledge, we are not aware of any open-resource PTQ method specifically for LIC models. For fair comparison, we implement the Range-Adaptive Quantization (RAQ) (Hong et al.,

---

[2]https://github.com/InterDigitalInc/CompressAI
[3]https://github.com/lumingzzz/TinyLIC

Table 1: BD-rate loss over floating-point models

| Model | Lu2022 | | | Cheng2020 | | Minnen2018 | |
|-------|--------|-------|-------|-----------|-------|------------|-------|
| Method | Ours | FQ-ViT | RAQ | Ours | RAQ | Ours | RAQ |
| Kodak | 3.70% | 7.06% | 29.40% | 4.88% | 27.84% | 5.84% | 30.41% |
| Tecnick | 6.21% | 12.13% | 31.04% | 6.86% | 29.95% | 8.23% | 31.55% |

2020) originally requiring model retraining as a PTQ approach; On the other hand, we also include the FQ-ViT (Lin et al., 2022) for comparative study. It is a PTQ method originally designed for image classification and objective detection using Transformer backbone. Here we extend its main idea to support the compression task. Besides, we compared MSE optimization and two methods (Sun et al., 2020; 2021) with retraining in the ablation experiment, and the more details are provided in the supplementary material.

**Testing Datasets.** Two popular datasets that contain diverse images are used for evaluation, i.e., the Kodak dataset with the image size of $768 \times 512$, Tecnick dataset with the image size of $1200 \times 1200$. The Peak Signal-to-Noise Ratio (PSNR) measures the image quality and bits per pixel (bpp) reports the consumption of compressed bitrate.

## 5.2 EVALUATION

**Quantitative Performance.** We plot R-D curves in Fig. 8 of supplementary material for various floating-point LIC models and their quantized INT8 counterparts, and further report the BD-rate performance (Bjontegaard, 2001) over 32-bit floating-point models for different PTQ approaches in Table 1. In the meantime, compression efficiency using both VVC Intra (VTM) and HEVC Intra (BPG) are also exemplified to help understand the relative performance gaps between quantized LIC and traditional image coder.

As seen, quantized INT8 LICs using our RDO-PTQ provide the least BD-rate loss to corresponding floating-point LIC models, greatly outperforming other PTQ alternatives (see averaged results in Table 1 and curves in Fig. 8). More importantly, having the anchor of Lu2022 [FP32], Lu2022 [INT8] using our RDO-PTQ provides similar performance as the VVC Intra on Kodak dataset, and largely outperforms it on Tecnick dataset.

As visualized in Fig. 8, the BD-rate loss enlarges at higher bitrates for the proposed RDO-PTQ on various floating-point LICs. One potential reason is the increase of channels for high bitrate LIC models (e.g., from 192 to 320). Note that we optimize the PTQ channel-wisely, accumulated quantization error is typical larger for the model with more channels. Such problem can be possibly resolved by optimizing the PTQ for all layers at once which however may need excessive amount of computations. This is an interesting topic for further study.

We notice that Sun et al. (Sun et al., 2022) also introduced a PTQ method to quantize LICs recently. They demonstrated that the influence of quantization error to the final reconstruction was different across channels. As a result, they manually categorized channels whose quantization may lead to larger reconstruction error into different groups. Although they included the reconstruction error instead of the quantization error into the discussion, they did not connect reconstruction performance with the weight, bias, and activation following the R-D optimization means, which is very different from our proposed RDO-PTQ. Unfortunately, we do not find any publicly accessible material for us to perform the comprehensive comparison.

## 5.3 ABLATION STUDIES

**Dynamic Range.** Previous methods usually tend to clip large parameter values to control the dynamic range, while our proposed RDO-PTQ, by contrast, optimizes the dynamics of tensor for better R-D performance. Without any publicly accessible resources, we manually reproduced the two SOTA methods (Sun et al., 2020; 2021) for comparison. More details of the experiment can be found in the supplementary materials. Fig.4 shows the result of weight only quantization. Fig. 5 shows the result of quantizing weight and activation.

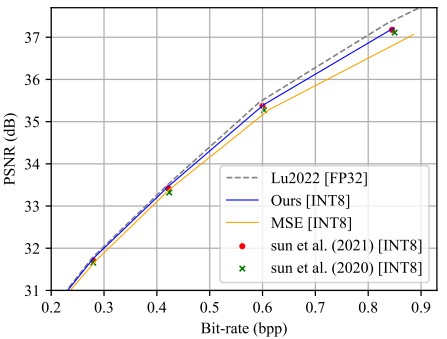

Figure 4: Weight only quantization.

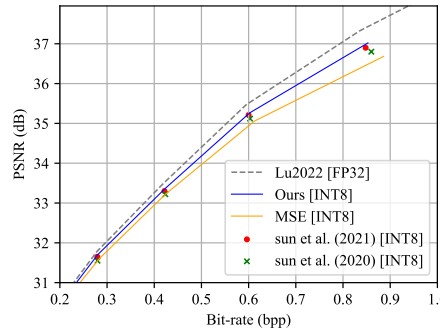

Figure 5: Weight and activation quantization.

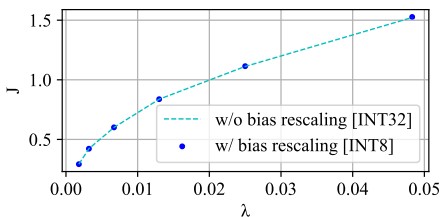

Figure 6: Effect of INT8 bias compared with INT32 bias.

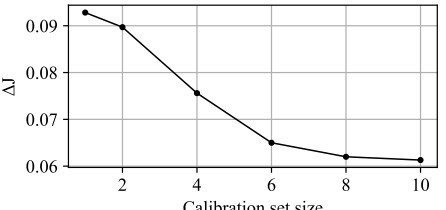

Figure 7: Comparison of RD as calibration set size increases.

Noted that Sun et al. (2020; 2021) only quantize weights in their work, as a result, we add the quantization of activations channel-wisely for fair comparison. The BD-rate losses over Lu2022 are $4.43\%$ and $6.78\%$ respectively which are higher than ours. Besides, the results illustrate the sub-optimality of MSE optimization.

**Bias Rescaling.** Taking INT32 bias without rescaling as the reference, we test the effect of bias rescaling on Lu2022. The common Min-Max quantization is adopted without quantizing activation, which does not require the calibration set and does not introduce quantization error of activation, which generally simplifies the problem for discussion. We test performance on Kodak dataset. As shown in Fig. 6, bias rescaling for fully 8-bit processing has no negative impact on performance.

**The Size of Calibration Set.** We randomly select images from ImageNet to formulate the calibration set for quantization factor determination in our RDO-PTQ. As shown in Fig. 7, when the size of calibration set increases to 10, there is basically no loss to the R-D performance compared with the floating point model. Therefore, we adopt 10 images for calibration in this work.

## 6 CONCLUSION

This paper studied the PTQ to directly quantize off-the-shelf, pretrained floating-point image compression models for their fixed-point counterparts. To retain the compression efficiency as the native floating-point LICs, we suggested the R-D optimized PTQ which was first justified theoretically and then proved experimentally. In RDO-PTQ, we determined the proper ranges of weight and activation channel-wisely from one layer to another, and layer-wise ranges of bias for subsequent rescaling, to which a tiny set of images (e.g., about 10) were sufficient to calibrate these model quantization related factors. Results revealed the superior efficiency of the proposed method, presenting large performance improvements to existing approaches. More importantly, our method did not require any model retraining, and offered a push-bottom solution for all existing LICs. An interesting topic for further study is to reduce the quantization loss at high bitrates.

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
