# OpenReview forum: "Rate-Distortion Optimized Post-Training Quantization for Learned Image Compression"
_ICLR.cc/2023/Conference — Submitted to ICLR 2023_

### Official Review · Reviewer_AqaT · 2022-10-24

**Confidence:** 4
**Correctness:** 3
**Technical Novelty And Significance:** 2
**Empirical Novelty And Significance:** 1
**Recommendation:** 5

**Clarity, Quality, Novelty And Reproducibility:**

Clear presentation of the paper. But not novel enough. Reproducibility is good since the proposed method is not difficult.

**Details Of Ethics Concerns:**

Refer to Strengths and weaknesses part

**Strength And Weaknesses:**

Strength:
1.	The proposed post-training quantization (PTQ) method can transform the model from Float32 format to Int8 precision without large-scale model retraining.
2.	Some theoretical discussions are provided to illustrate the motivation of the proposed method.
3.	The experimental results demonstrate the effectiveness of the PTQ method in neural compression models, i.e., tolerable performance drop but much more light-weighted neural network.
Weakness:
1.	The only difference between using PTQ in LIC models and conventional neural networks is the format of task loss. This paper simply applies the PTQ method for the task of LIC, thus makes somewhat trivial technical contributions.
2.	The theoretical derivations and the relevant conclusions in this paper have been given in the paper of AdaRound [1].
3.	The setting of this paper is to transform the neural network from FP32 to INT8. But actually this setting is defined by the author without any explanations. Some other choices can be adopted, e.g., INT4 or unsigned INT16.
4.	There are some vague claims without evidence, such as, ‘Such inappropriate processing of bias may lead to catastrophic results. For example, having the bias in INT32 precision may cause data overflow of the INT32 accumulator; while setting bias as zero would degrade the model performance significantly.’
References:
[1] Up or down? adaptive rounding for post-training quantization. Nagel et al., ICML 2020.



**Summary Of The Paper:**

This paper suggests the use of post-training quantization to quantize neural image compression models, where the model parameters are compressed from Float32 format to Int8 precision. The weight, bias and activation parameters are processed with post-training optimization. Some theoretical justifications are provided to prove that the quantization error is not monotonically related to the rate-distortion performance. The experimental results show the proposed model quantization method can convert the network precision with only 3.70% BD-rate increases.

**Summary Of The Review:**

The PTQ method has been applied on other tasks for many times. This paper simply applies the PTQ method into learned image compression models. The theoretical analyses in this paper also follow previous papers without new conclusions.

---

> ### Author Response · Authors · 2022-11-18
> **Response to Reviewer AqaT**
>
> ## Technical contribution
> Learned image compression (LIC) is a multi-loss task, in which we should not only ensure the quality of the reconstructed output but also take care of the consumed bits. It is a rate-distortion optimization (RDO) problem, which is largely different from existing vision tasks (e.g., image super-resolution, image classification, object detection) that solely examine the reconstructed quality (MSE) or accuracy (cross-entropy) considered. It motivates us to quantize LICs using rate distortion criteria, which is the main contribution that has never been explored for compression model quantization. The modified rounding mechanism and dynamic range determination adapted to such a compression task are purposely dedicated to fulfilling the RDO.
>
> ## Comparison to AdaRound
> In optimization problems, it is common to conduct loss analysis through Taylor expansion, and we use a similar method to illustrate our theory.
>
> AdaRound is a rounding mechanism where the author transforms the rounding-to-nearest to floor or ceil rounding through Taylor expansion and extra boundary conditions. AdaRound works well on high-level tasks that only consider accuracy, by using the individual MSE metric to reduce the quantization loss. This is very different from our approach which uses  the bitrate and MSE jointly to balance the rate and distortion in compression tasks.
>
> ## Why prefer INT8
> INT8 has been widely used in applications, thus we choose INT8 as our target in the main paper. In addition, in the supplementary materials (see Section E), we also show the results under 10-bit (INT10). When the bit width is 10-bit or higher, the loss is negligible. In the future, we will try even lower bit-width. In the supplementary materials (see Section A.2), we have added more details and explanations.
>
> ## Vague claims on bias
> We have added reference papers and explanations about bias quantization in supplementary materials (see Section A.3) which complement to our statement with more details.

---

> ### Comment · Reviewer_AqaT · 2022-12-04
> **thanks for the feedback**
>
> thanks for the feedback, some of the concerns have been addressed. It's still difficult to agree that using PTQ into image compression can be regarded as a distinguished novel contribution.  Adjusting the scale of parameters may be a useful method. However, this technique is also a general technique which should be able to be applied into many tasks. No matter in the image restoration task or in the image compression task, the PTQ trick is applicable.

---

> > ### Author Response · Authors · 2022-12-04
> > **Response to Reviewer AqaT**
> >
> > Thanks for your response. We would like to clarify that we do not claim "the novelty of using PTQ into image compression'' as mentioned by you. Instead, our major contribution is the R-D optimized PTQ (RDO-PTQ) used for LIC since existing works mainly apply the MSE-based PTQ.  This is because, as for the compression task, the distortion D and bit rate R shall be considered jointly, instead of the distortion (MSE) only.
> >
> > This work not only theoretically justifies that  the quantization of the compression model using MSE loss between the quantized and the original floating-point weights/activations is sub-optimal but also reports the effectiveness of our RDO-PTQ method through comprehensive  experiments and ablation studies.
> >
> > Besides, in this revised submission, two typical QAT methods (e.g., [1], [2]) are provided in Fig. 4 and Fig. 5. As you can see, our RDO-PTQ achieves even better performance than the QAT method that requires the retraining. Instead,  our method is a plug-and-play approach for any off-the-shelf LIC models. This work offers a novel solution from a rate-distortion optimization perspective.
> >
> > ---
> >
> > [1] Sun, Heming, et al. "End-to-end learned image compression with fixed point weight quantization." 2020 IEEE International Conference on Image Processing (ICIP). IEEE, 2020.
> >
> > [2] Sun, Heming, Lu Yu, and Jiro Katto. "Learned Image Compression with Fixed-point Arithmetic." 2021 Picture Coding Symposium (PCS). IEEE, 2021.

---

### Official Review · Reviewer_ra2Y · 2022-10-31

**Confidence:** 4
**Correctness:** 3
**Technical Novelty And Significance:** 3
**Empirical Novelty And Significance:** 2
**Recommendation:** 8

**Clarity, Quality, Novelty And Reproducibility:**

The paper is clear and well organized. The idea of the proposed contribution seems interesting. Indeed, the proposed method needs to compress few images to optimize the transformation of weight, bias, and activation of underlying LIC model from its native 32-bit floating-point (FP32) format to 8-bit fixed-point (INT8) precision for fixed-point inference onwards. An algorithm is provided to illustrate the different steps constituting the proposed method.



**Strength And Weaknesses:**

Strength :
The paper is clear and well organized. The idea of the proposed contribution seems interesting. Indeed, the proposed method needs to compress few images to optimize the transformation of weight, bias, and activation of underlying LIC model from its native 32-bit floating-point (FP32) format to 8-bit fixed-point (INT8) precision for fixed-point inference onwards. In addition, the experimental results have shown the effectiveness of the proposed method.

Weaknesses :
Other state-of-the-art methods could be considered in the comparison.

**Summary Of The Paper:**

This paper studies the use of Post-Training Quantization (PTQ) to directly process pretrained, off-the-shelf Learned Image Compression  (LIC) models.  The proposed study proves theoretically that minimizing the mean squared error (MSE) in PTQ is suboptimal for compression task, and proposes a novel Rate-Distortion (R-D) Optimized PTQ (RDO-PTQ) to best retain the compression performance.

**Summary Of The Review:**

The paper is clear and well organized. The proposed method is well described and the experimental results have shown their effectiveness. The proposed contribution is interesting.

---

> ### Author Response · Authors · 2022-11-18
> **Response to Reviewer ra2Y**
>
> Thank you for your constructive suggestions. We have added an extra experiment comparing with two SOTA quantization methods (e.g., [1], [2]). Unfortunately, there is no public code available for reference in these two papers, we faithfully reproduce them by ourselves and conduct the evaluation in the ablation experiment of the new submission (see Sec. 5.3). Additionally, we also report the runtime and  other implementation details in the supplemental materials for better reference.
>
> ---
>
> [1] Sun, Heming, et al. "End-to-end learned image compression with fixed point weight quantization." 2020 IEEE International Conference on Image Processing (ICIP). IEEE, 2020.
>
> [2] Sun, Heming, Lu Yu, and Jiro Katto. "Learned Image Compression with Fixed-point Arithmetic." 2021 Picture Coding Symposium (PCS). IEEE, 2021.

---

### Official Review · Reviewer_soh4 · 2022-11-01

**Confidence:** 3
**Correctness:** 3
**Technical Novelty And Significance:** 2
**Empirical Novelty And Significance:** 2
**Recommendation:** 3

**Clarity, Quality, Novelty And Reproducibility:**

Overall the paper is well written and easy to follow.
Minor suggestion:
1) Figure 4 at its current state is very difficult to visualize since there are many lines overlapped with each other in a very small region. I would advice the authors to try out alternate ways (maybe changing the axis scale or bar plot instead of line plot) to make the plots more clear.
2) In the 3rd paragraph of the introduction, the line "QAT method requires model re-training" seems to be wrong, and might require correction. I think the authors wanted to refer 'PTQ with fine-tune' methods instead of 'QAT'.
3) In section 4.1, "Assuming a trained compression task model with R-D metric $J(x,w)$" in place of "Assuming a trained compression task model $J(x,w)$".

**Strength And Weaknesses:**

Pros:
The idea of utilizing the RD term as an objective function to find the optimum scaling factor in the context of LIC task seems novel and the main contribution of the paper.

Cons:
I have several concerns about the papers as outlined below:

Novelty and Prior-works: The authors claim that applying PTQ for LIC task is one of their main contributions. However, there have been several prior works that have already applied PTQ on LIC tasks. See [1-4]. The authors refer [1], and [2] in the introduction section as a quantization aware training (QAT) method, which I believe, is not a correct statement. Rather these methods are PTQ followed by fine-tuning (which is very common in computer vision tasks, specially for bit-width lower than 8), similar to [4]. Other than that, the work in [3] have directly applied existing PTQ method on LIC task without requiring any fine-tuning, similar to this work. Although the authors refer [3] in the introduction section as a part of their motivation (cross-platform inconsistency issue of the floating-point LIC methods), they do not discuss how the proposed work is different from theirs. I believe the main claim of this work should be 'PTQ on LIC without requiring any fine-tuning' (if we do not consider [3], of course).

Significance: The authors claim that the quantization error and compression metric do not have any monotonic relation (which is the main motivation behind their proposed loss). However, from Fig. 2, this claim seems to be valid only at the edge of minimum R-D loss(i.e., $\Delta$J ~= 0) while they follow a monotonic relationship everywhere else. There was no theoretical justification or any insightful discussion behind such observation. This also questions the significance of using the R-D loss as the objective function rather than simple MSE loss as the authors in [3] showed pretty impressive results on the same benchmark (Fig 5(a) in [3]) with MSE loss only.

Experiments: Comparison with other PTQ methods are missing. While [3] and [4] might be considered as pretty recent works, at least comparison with [1] and [2] is desirable. Also, in the ablation study, there could be an additional experiment (a graph similar to Fig 4) that clearly shows the performance gain of R-D loss over MSE loss where everything else (except the objective function) is same for fair comparison.

1. Sun, Heming, et al. "End-to-end learned image compression with fixed point weight quantization." 2020 IEEE International Conference on Image Processing (ICIP). IEEE, 2020.
2. Sun, Heming, Lu Yu, and Jiro Katto. "Learned Image Compression with Fixed-point Arithmetic." 2021 Picture Coding Symposium (PCS). IEEE, 2021.
3. He, Dailan, et al. "Post-Training Quantization for Cross-Platform Learned Image Compression." arXiv preprint arXiv:2202.07513 (2022).
4. Sun, Heming, Lu Yu, and Jiro Katto. "Q-LIC: Quantizing Learned Image Compression with Channel Splitting." arXiv preprint arXiv:2205.14510 (2022).


**Summary Of The Paper:**

This work proposes a framework for applying off-the-shelf post-training quantization (PTQ) methods for learned image compression (LIC) task. The authors show that the optimum quantization scaling factor may not be discovered by only minimizing the MSE loss for LIC task and provide theoretical insights on this. To alleviate this, they propose to use a rate-distortion (RD) loss as their objective function where both the distortion and bit-rate are coupled by a Lagrange multiplier. Their experiments on 2 benchmark datasets show that INT8 quantized LIC (using their PTQ method) can achieve similar RD performance compared to the original 32-bit counterparts while outperforming the other quantization baselines.

**Summary Of The Review:**

Overall, due to the limited novelty, lack of proper discussion about the related works, lack of proper experiments and justification on why R-D loss is a better objective function compared to the MSE loss, I can not recommend for acceptance. However, I am willing to change my opinion if the authors can provide solid answers to my above concerns.

---

> ### Author Response · Authors · 2022-11-18
> **Response to Reviewer soh4**
>
> ## Definition and figure clarity
> Thanks for your suggestions.  We have modified the relevant expressions (see Section 1 and Section 4.1) and used Table 1 to replace the previous RD curves. The original RD curves are part of supplementary materials (see Section A.4).
>
> Unfortunately, we respectfully disagree with your comments regarding the QAT and PTQ definitions.
>
> For the QAT, the model parameter (weight, activation, etc) is updated in backpropagation during the retraining (or the fine-tuning as you said) to a level that is better for quantization, which practically changes the  parameters of the original floating-point model. This is why it is so-called quantization-aware training (QAT).  Please refer to Algorithm 1 in [2] where the parameters of the original floating model are adapted. The same process is used in [1] as well. That is why we categorize them into the QAT method.
>
> On the contrary, as for the PTQ, the parameters of the original floating point model are fixed for model quantization, for which we can only adapt the scaling factor accordingly to quantize the floating parameters (but not change the original floating parameters in QAT).
>
> As also suggested by another reviewer, we are not arguing that the PTQ is definitely better than the QAT. This work just offers a solution to point out that RDO-PTQ is vital for compression tasks. With the proposed RDO-PTQ, we can not only save runtime but also provide a similar or even better performance than the QAT methods used in the comparison.
>
> ## Monotonic relation
> We believe that the claims here are biased and unfair to the authors.
>
> > **Localized Non-monotonic Behavior.** In the meantime, we visualize the absolute  quantization error of weight, e.g., $\Delta \boldsymbol{w}$ with compression task loss measured by the $\Delta J$, which further confirms the theoretical justification aforementioned. As seen in Fig. 2, although over a wide range of $\Delta \boldsymbol{w}$, it is monotonically related with the R-D loss $\Delta J$ in compression task; it presents localized non-monotonic behavior where the minimization of quantization error does not lead to the minimum of $\Delta J$, which, consequently, degrades the overall compression performance of underlying LIC model if we pursue the minimization of quantization error for model quantization.
>
> First of all, in the  Sec. 4.1 of the original submission, we have clearly stated the problem and copied it here for your reference. As you can see, we have never claimed "do not have any monotonic relation", instead we have given a paragraph discussing so-called localized non-monotonic behavior.
>
> Furthermore, as in Sec. 4.1, we have clearly provided the theoretical justification for why the MSE loss-based PTQ quantization is sub-optimal, which is not the case as claimed like "there was no theoretical justification ....".
>
> ## MSE loss
> In reference [3], it is not just using an MSE loss as  stated in Sec. 3.1 of the [3]. Instead, it first uses the MSE loss to derive the scaling factors and then applies the per-block fine-tuning.
>
> ## Comparison to prior works
> We add some contrast with some prior works [1,2] and MSE optimization. As there is no any public code for reference in the current LIC quantization papers, we try to reproduce them. According to the papers, they only quantize the weight of the model. Thus we apply channel-wise activation quantization for fair comparison. The results are presented in ablation experiment (see Fig. 4 and Fig. 5). Detailed experimental settings can be found in the supplementary materials (see Section C).
>
> ---
>
> [1] Sun, Heming, et al. "End-to-end learned image compression with fixed point weight quantization." 2020 IEEE International Conference on Image Processing (ICIP). IEEE, 2020.
>
> [2] Sun, Heming, Lu Yu, and Jiro Katto. "Learned Image Compression with Fixed-point Arithmetic." 2021 Picture Coding Symposium (PCS). IEEE, 2021.
>
> [3] He, Dailan, et al. "Post-Training Quantization for Cross-Platform Learned Image Compression." arXiv preprint arXiv:2202.07513 (2022).

---

> > ### Comment · Reviewer_soh4 · 2022-11-29
> > **Response to the authors**
> >
> > We would like to thank the authors for the corrections made in the figure, text, and most importantly, their follow-up clarification on the PTQ and QAT definition. We would love to see this clarification in the final version of the manuscript if accepted.
> >
> > However, our main concerns are still not addressed we believe. Let us clarify about it:
> >
> > **Followup on Monotonic Relation**: We agree that the authors indeed mentioned about the 'localized' behavior of the non-monotonic relationship and it was our mistake not to mention about it in our original review. However, the authors might have misunderstood our main point, which is, the reason or any insightful discussion behind such highly **localized behavior** was missing in the paper. We agree with the authors that they have provided the theoretical justification for why the MSE loss-based PTQ quantization is sub-optimal (by showing  the dependency of R-D loss on second order derivative of $J$), however my concern was that there was no theoretical justification on such localized behavior shown in Fig. 2. The authors mention that *"it presents localized non-monotonic behavior where the minimization of quantization error does not lead to the minimum of $\Delta J$, which, consequently, degrades the overall compression performance of underlying LIC model"*, however my concern was whether there would be any **significant degradation** as it is fairly clear from Fig. 2 that minimization of quantization error indeed leads to **very close** to minimum of $\Delta J$.
> >
> > **Followup on the new experiments**: We thank the authors for additional experiments and comparison with [1] and [2]. However, we have few confusions about them: 1) is there any particular reason for comparing with these methods on this newly introduced DIV2K dataset instead of the original Kodak and Technick datasets? It is also kind of misleading because there is no such mention about this newly introduced dataset in the main paper. 2) The authors only mention: "*The BD-rate losses over Lu2022 are 4.43% and 6.78% respectively which are higher than ours*", which is really confusing. There are two methods here evaluated on two different cases, weights only (fig 4) and weight+activation (fig 5). Which figure/experiment are the authors referring to for this aforementioned result? Which methods are they referring to? What is the actual improvement (in numeric) over the proposed method? From our visual inspection from fig 4 and 5, it seems the method proposed by Sun et al. (2021) has almost similar performance compared to the proposed method (for both weight-only and weight+activation case), which again brings back to my original point: how significant this R-D loss would be over the existing MSE-based methods.

---

> > > ### Author Response · Authors · 2022-12-01
> > > **Response to Reviewer soh4**
> > >
> > > Thanks for your response. We further clarify them for better presentation and for your understanding.
> > >
> > > ## Localized non-monotonic behavior
> > >
> > > As seen from Eq. 8, $\Delta J$ is a multi-variable function,
> > >
> > >    - with the increase of quantization error $\Delta w$, R-D loss $\Delta J$ is mainly determined by the square term $\Delta w^2$, and the effect of the first order term $\Delta w$ is negligible. This is the global monotonic behavior of  $\Delta J\approx\Delta w^2$ as in Fig. 2.
> > >
> > >    - However when $\Delta w$ is small, the impact is compound of $\Delta w^2$ and  $\Delta w$ (and so does the activation error) as in Eq. 8, making the localized non-monotonic behavior.
> > >
> > >  As $\Delta w$ is a multidimensional tensor, different elements may have different $\Delta w$. For example, some elements may have small $\Delta w$s reporting clustered non-monotonic relations; while some elements may have large $\Delta w$ leading to the global monotonic behavior.
> > > We believe, by connecting Eq. 8 and Fig. 2, you can have a more clear idea about such localized behavior.
> > >
> > > It should be noted that the result in Fig. 2 is obtained only after the first convolutional layer in the main encoder. It is for visualization purposes. By quantizing more layers, errors will be accumulated.  From the final compression results in Fig. 4 and Fig. 5 for only MSE and R-D optimization, we can clearly observe that the MSE-only optimization degrades the overall performance more, especially at high bitrates. This is because, at high bitrates, more non-zero elements in activations are quantized leading to larger performance loss.
> > >
> > > Thus Fig. 2 is just used as a visualization of localized non-monotonic behavior. You need to compare the results in Fig. 4 and 5 to understand the quantization impact. We believe you will agree that MSE-only optimization indeed brings significant degradation of compression performance.
> > >
> > > We will make the presentation clearer in the final submission following your suggestion.
> > >
> > > ## New experiments
> > >
> > > First, we really appreciate your understanding of our explanation of QAT and PTQ in the last round.
> > >
> > > 1. [1,2] are QAT methods, which require the re-train to update the floating-point model. We use the same DIV2K as the training set; while results in both Fig. 4 and Fig. 5 are tested on Kodak.
> > >
> > > 2. As [1,2] only quantize weights in their works, as a result, we add the quantization of activations channel-wisely for a fair comparison.   The BD-rate losses (e.g., 4.43\% and 6.78\%) mentioned in Sec. 5.3 is obtained under weight + activation quantization (in Fig.5) by comparing with the original floating-point model Lu2022, which is larger than ours (e.g., 3.70\%) in Table 1.
> > > Alternatively, we have weight-only quantization in Fig. 4, BD-rate loss are 1.58\%, 1.94\%, 3.57\% for ours, [2], [1], which shows that our method is also better.
> > >
> > > &emsp;&emsp;We apologize for the vague presentation and will make it clear in the revision.
> > >
> > > 3. Compared with [2], the BD-rate loss is from 4.43\% to 3.70\% (see Table 1 and Fig. 5), which can not be simply neglected.  On the other hand, we want to point out that [2] is a QAT approach while ours is a PTQ method, which is fundamentally different.  As always, we are not discouraging the use of MSE-only quantization optimization or QAT method, this work is just offering a new solution from an RDO-PTQ perspective. And more importantly, theoretically and experimentally we show that RDO-PTQ is an attractive push-bottom solution for quantizing the LICs.

---

> > > > ### Comment · Reviewer_soh4 · 2022-12-01
> > > > **A Quick followup**
> > > >
> > > > Thank you for the response. I just have a quick question as I am really confused about the new explanation:
> > > > The authors mention in the follow-up response that the global monotonic behavior of $\Delta J \approx \Delta w^2$ as in Fig. 2. However, as far as I can see from Fig. 2, most of the monotonic region in Fig. 2 is covered by $\Delta w < 1$; while the authors mention:  *$\Delta J$ is mainly determined by the square term $\Delta w^2$.* However, for $\Delta w < 1$, shouldn't the loss term be determined mostly by $\Delta w$ as $\Delta w^2 << \Delta w$ when $\Delta w<1$? The same reasoning applies to the authors' 2nd point as well, the compound dependency should go away as $\Delta w$ becomes even smaller and close to $0$. Based on my above reasonings and the plot in Fig. 2, can the authors please clarify how their newly presented arguments still hold?

---

> > > > > ### Author Response · Authors · 2022-12-04
> > > > > **Response to Reviewer soh4**
> > > > >
> > > > > Thank you again for your comment.
> > > > > We further explain why non-monotonic behavior is more obvious when $\Delta \boldsymbol{w}$ is small.
> > > > >
> > > > > In Fig. 2, we assume the fixed $\Delta {x}$ to study the relationship between $\Delta J$ and $\Delta {w}$.  For visualization purposes, we actually plot  $\Delta J\sim$  $|\Delta {w}|$.
> > > > >
> > > > > As revealed by extensive experiments, we observe that $\frac{\partial ^2 J}{\partial \boldsymbol{x} \partial \boldsymbol{w}}$ is more than two orders of magnitude smaller (i.e., $< 10^{-2}\times$ smaller ) than $\frac{\partial ^2 J}{\partial \boldsymbol{w}^2}$, e.g., $\frac{\partial ^2 J}{\partial \boldsymbol{w}^2} > 100 \times$ $\frac{\partial ^2 J}{\partial \boldsymbol{x} \partial \boldsymbol{w}}$. Recalling the presentation in Eq. 8,  only when $\Delta \boldsymbol{w}$ is small enough (e.g., at the order of magnitude of $< 10^{-2}$),  $\frac{\partial ^2 J}{\partial \boldsymbol{x} \partial \boldsymbol{w}} \Delta \boldsymbol{x} \Delta \boldsymbol{w}$ and $\frac{\partial ^2 J}{\partial \boldsymbol{w}^2} \Delta \boldsymbol{w}^2$ are about the same scale, which then leads to the localized behavior as plotted in Fig. 2. This is because  elements in $\Delta \boldsymbol{w}$ can be negative, zero, and positive after the quantization, making the sum of $\frac{2 \partial ^2 J}{\partial \boldsymbol{x} \partial \boldsymbol{w}} \Delta \boldsymbol{x} \Delta \boldsymbol{w}$ +  $\frac{\partial ^2 J}{\partial \boldsymbol{w}^2} \Delta \boldsymbol{w}^2$ in Eq. 8 fluctuate non-monotonically as visualized in zoom-in part of Fig. 2.
> > > > >
> > > > > For global monotonic behavior, we clarify the previous vague statement,
> > > > > - with the increase of $\Delta \boldsymbol{w}$, $\Delta J$ is mainly determined by the square term $\Delta \boldsymbol{w}^2$, and the effect of the first order term $\Delta \boldsymbol{w}$ is negligible. This is the global monotonic behavior of $\Delta J\propto\Delta \boldsymbol{w}^2$ (instead of $\Delta J \approx \Delta \boldsymbol{w}^2$) as in Fig. 2.

---

### Official Review · Reviewer_LeDd · 2022-11-03

**Confidence:** 4
**Correctness:** 4
**Technical Novelty And Significance:** 2
**Empirical Novelty And Significance:** 2
**Recommendation:** 5

**Clarity, Quality, Novelty And Reproducibility:**

Overall, the clarity and quality of the paper is very high. I appreciated the theoretical explanation of the suboptimality of minimizing mse for quantization. The notation and various terms in Section 4.2 and 4.3 (where the core algorithm is explained) could be somewhat clearer, but it was understandable with a careful reading.

While I don't know of another paper that explicitly does RDO-PTQ, novelty still seems fairly low as described above.

I have no concerns about reproducibility, though it's always best to provide open source code.

**Strength And Weaknesses:**

Strengths:
1. The paper addresses an important, practical challenge for learned image compression: decode reliability and reducing space/time complexity.
2. The method is general (can be applied to any model) and appears to outperform other PTQ approaches.
3. The evaluation uses multiple datasets and multiple models, which minimizes concerns that it may not be widely applicable in practice. Also, one of the models is quite recent (Lu 2022).

Weaknesses:
1. Although post-training quantization is convenient, fine-tuning is not a major barrier. So additional evaluation showing the benefit (or lack thereof) of fine-tuning would strengthen the paper. In practical scenarios, additional training time is negligible compared to the lifetime of the codec and potential RD benefits.
2. The central insight is not very surprising or novel. Although RDO-PTQ outperforms the baseline methods, the basic expectation in compression is that RDO should always be performed and will always be best when it is possible/practical to apply.

**Summary Of The Paper:**

This paper presents a method for post-training quantization (PTQ) for learned image compression models. Quantization offers three main benefits: (1) fixed-point math is deterministic, which ensures reliable decoding (this is not guaranteed when different hardware is used to decode compressed representations using floating-point operations), (2) it reduces space complexity if int8 weights are used instead of float32, and (3) it reduces time complexity since typical parallel hardware can perform int8 operations more quickly than float32 or float16. Furthermore, PTQ occurs after training, as the name implies, and does NOT require additional fine-tuning to achieve good rate-distortion performance relative to the original float32 model. This simplifies application and reduces total training time.

The authors show, both theoretically and empirically, that selecting quantization parameters that minimize mean-squared error (mse) to the float parameters does NOT maximize rate-distortion performance. To address this, they propose rate-distortion optimization (RDO) for the quantization parameters and describe an algorithm for doing this in detail.

The empirical evaluation shows a relatively small drop in RD performance using RDO-PTQ. In particular, the drop is smaller (i.e. the resulting RD performance is better) than two previous methods (range-adaptive quantization (RAQ) and FQ-ViT). The authors run the evaluation using two image datasets and three different image compression models.



**Summary Of The Review:**

This paper addresses a practical problem for learned image compression, and the proposed algorithm outperforms the two baseline methods included in the empirical evaluation. The method, RDO-PTQ, is well-motivated and has many nice properties: it is generally applicable, relatively simple, does not require additional fine-tuning, and leads to only a relatively small drop in rate-distortion performance.

As discussed earlier, understanding the potential benefit of fine-tuning would strengthen the paper. I also think the outcome is more or less expected (RDO should always outperform a proxy loss for compression models unless it can be shown that the proxy is identical), and the applied nature of the result is interesting to learned image compression practitioners but is not a great fit for ICLR in general.

---

> ### Author Response · Authors · 2022-11-18
> **Response to Reviewer LeDd**
>
> ## Additional evaluation
> We add two typical QAT methods (e.g., [1], [2]) which require the model retraining in  Fig. 4 and Fig. 5. More details related to the experimental setup (e.g., used hardware, processing steps) and running time are added in the supplementary materials (see Section C) due to the space limitation.
>
> We agree with the reviewer that  additional re-training time might be  negligible for a codec in use.  This work offers the PTQ method as a potential solution besides the QAT approaches. As you can observe, the proposed RDO-PTQ method not only saves the runtime but also provides a similar or even better performance than the QAT methods in literature (see Fig. 4 and Fig. 5), not to mention that our method is a push-bottom solution that is attractive for practitioners.
>
> ## The novelty of RDO-PTQ method
> To the best of our knowledge, it is the first work to introduce the RDO strategy for model quantization. The success of the use of RDO in compression tasks does motivate this work.  Prior to our work, model quantization mainly uses the MSE loss under the assumption that quantizing the parameters with the least error is sufficient.  However, this work not only theoretically clarifies the quantization of the compression model using MSE loss between the quantized and the original floating-point weights/activations is sub-optimal but also proves the effectiveness of our RDO-PTQ method through comprehensive comparison experiments and ablation studies.
>
> ---
>
> [1] Sun, Heming, et al. "End-to-end learned image compression with fixed point weight quantization." 2020 IEEE International Conference on Image Processing (ICIP). IEEE, 2020.
>
> [2] Sun, Heming, Lu Yu, and Jiro Katto. "Learned Image Compression with Fixed-point Arithmetic." 2021 Picture Coding Symposium (PCS). IEEE, 2021.

---

### Official Review · Reviewer_8wCS · 2022-11-03

**Confidence:** 4
**Correctness:** 3
**Technical Novelty And Significance:** 2
**Empirical Novelty And Significance:** 2
**Recommendation:** 3

**Clarity, Quality, Novelty And Reproducibility:**

Quality: Due to the lack of novelty and experimental results, I am afraid the quality is not good.

Clarity: The paper is easy to follow in general.

Novelty: The novelty is really margin.

Reproducibility: The paper is reproducible since Adaround has been already imported in some model compression libraries such as AIMET.

**Strength And Weaknesses:**

Strength

The paper is one of very few works which studied the network quantization for learned codec.

Weaknesses

1. The novelty is really margin. AdaRound and dynamic range determination have been proposed in previous works. I wonder what is new when you utilize AdaRound in the specific task (i.e. learned codec)?
2. The writing can be improved. In Section 4.1, the equations are quite similar with the ones in AdaRound (Section 2. Motivation). Section 4.2 looks like some common equations for rate-distortion loss. Section 4.3 just introduces the N (dynamic range scaling) for weight/bias/activation.
3. The experiments are not enough. There is no ablation studies for the final results. Now that you have three methods in detail, AdaRound for weight, dynamic range for weight and dynamic range for activation, which is the individual contribution to the final result?
4. There is no running time evaluation.
5. This paper applied the network quantization into a transformer-based learned codec. There is a related work which also quantized the transformer-based learned codec (MobileCodec), it is better to cite the paper and make a comparison.

    MMSys '22: MobileCodec: neural inter-frame video compression on mobile devices





**Summary Of The Paper:**

This paper applies PTQ in learned image compression. Two methods are exploited. One is AdaRound (ICML'20) for weight, and the other is a learnable dynamic range for both weight and activation.
My main concern is about the novelty of this paper. AdaRound was presented in ICML'20, and learnable dynamic (clipping) range has also been studied in quite a few of literatures such as PACT.

PACT: Parameterized Clipping Activation for Quantized Neural Networks

**Summary Of The Review:**

As described in the above weaknesses, the proposal looks like just a combination of some previous methods and some mathematical derivations are also quite close to the previous paper. Besides, the experimental results should be improved, there is no ablation study and the comparison with previous works can also be more comprehensive.

---

> ### Author Response · Authors · 2022-11-18
> **Response to Reviewer 8wCS**
>
> ## The particularity of Learned Image Compression (LIC)
> Learned image compression (LIC) is a multi-loss task, in which we should not only ensure the quality of the reconstructed output but also take care of the consumed bits. It is a rate-distortion optimization (RDO) problem, which is largely different from existing vision tasks (e.g., image super-resolution, image classification, object detection) that solely examine the reconstructed quality (MSE) or accuracy (cross-entropy) considered. It motivates us to quantize LICs using rate distortion criteria, which is the main contribution that has never been explored for compression model quantization. The modified rounding mechanism and dynamic range determination adapted to such a compression task are purposely dedicated to fulfilling the RDO.
>
> ## Comparison to AdaRound
> Thank you for your comment. First of all, it should be noted that our rounding mechanism is definitely different from the AdaRound. AdaRound works well on high-level tasks that only consider accuracy, by using the individual MSE metric to reduce the quantization loss.
> This is very different from our approach which uses  the bitrate and MSE jointly to balance the rate and distortion in compression tasks.
>
> Specifically, AdaRound replaces the ``rounding-to-nearest" with the "floor or ceil" operation.
> In brief, it formulates the quantization as a continuous optimization process based on soft quantization variables:
> $$
> \arg \min_{\boldsymbol{v}} \ \lVert \widehat{\boldsymbol{w}}\boldsymbol{x} - \boldsymbol{wx} \rVert^2 + \lambda f_{reg}(\boldsymbol{v}).
> $$
> The additional term $\lambda f_{reg}(\boldsymbol{v})$ is a differential regularization introduced to make the soft-quantized weights to converge towards floor or ceil rounding.
>
> In our method, we optimize the quantization parameters (e.g., scale factors) by calculating RD loss compared to the floating-point model:
> $$
>  s_{\boldsymbol{x}}, s_{\boldsymbol{w}} ,s_{\boldsymbol{b}}
>              = \arg \min_{ s_{\boldsymbol{x}}, s_{\boldsymbol{w}} ,s_{\boldsymbol{b}}} \ \lVert (\widehat{R} + \lambda \cdot \widehat{D}) - (R_0 + \lambda \cdot D_0) \rVert^2.
> $$
> As seen in equation, our R-D optimization would drive a more proper quantization towards the minimization of the R-D loss in the compression task, which is fundamentally different from the ''ceil'' or ''floor'' operation in AdaRound.
>
> ## Modification and additional details
> Thank you for your constructive comments. We have reorganized the paper to make our statements  more clear and more comprehensive. For example, we  add the comparison with only MSE loss and another two QAT methods that require model retraining in Section 5 (see Fig. 4 and Fig. 5) to further evidence the effectiveness of our method. The runtime is listed in the supplemental materials, reporting about 78\% reduction against the QAT models with re-training.
>
> Additionally, we have included the ``Mobile Codec'' as the reference in Related Work (Section 2). Unfortunately, it would take substantial efforts to reimplement the Mobile Codec for a fair comparison because their materials are not made publicly accessible yet, we would like to do it as a future study. More importantly, the method used in Mobile Codec belongs to the QAT category which is already compared in our submission.

---

### Official Review · Reviewer_6kPp · 2022-11-03

**Confidence:** 3
**Correctness:** 3
**Technical Novelty And Significance:** 2
**Empirical Novelty And Significance:** 2
**Recommendation:** 5

**Clarity, Quality, Novelty And Reproducibility:**

The main motivation is easy to follow, however the paper lacks in detailed description of the experiments and complete analysis (see weaknesses). The originality of the technical contribution (mainly modifying the loss function to include the compression rate to target image compression) is rather limited, therefore insightful and complete analysis would be important for the paper.

**Strength And Weaknesses:**

*Strengths*

- The loss due to the quantization is with the proposed method smaller than with other post-processing methods.
- The method is simple and works better than other methods shown.
- The motivation, to use the loss function during optimization, which is also used in evaluation, makes sense.

*Weaknesses*
- The main technical contribution of the work is considering the compression rate in addition to the MSE loss. However, the comparison to the baseline using only MSE is only shown in the supplement. In my opinion, this comparison and discussion should be part of the main paper.
- Some details related to the experimental setup are missing (e.g., algorithm runtime, needed to get the quantized model and the optimized scaling factors, used hardware, calibration set selection).
- The quantization parameters are obtained by training on a small calibration dataset. From the paper it is unclear how this set has been selected and how sensitive the results are to the selection, e.g., by measuring the variance of performance over different image sets. I am wondering whether the calibration set is dataset specific in the experiments, and if that is the case, how the performance would be with an unsuitable calibration set. This clarification is important to fulfill the claim, that is truly a plug-and-play method without model retraining.
- For completeness it would be interesting to compare also to a method which retrains the compression model, as the paper claims to be more efficient in terms of training time and practicability, e.g., by reporting the time needed.
- The algorithm needs to be run for every desired bit rate.


**Summary Of The Paper:**

The paper presents a method for quantizing floating-point representation to fixed-point precision in a learned image compression framework. The main advantage of the proposed method is, that it can be added to pretrained off-the-shelf compression models, making it useful in practice. The paper adapts an existing post-training quantization method to image compression task by considering the rate-distortion during optimization instead of only mean squared error. Results are shown by adding the approach to 3 existing methods.

**Summary Of The Review:**

The paper presents a simple, but apparently effective method, however the main weaknesses are in the experimental section and performance analysis. Without additional insights and description of the effect of the calibration dataset it is difficult to judge how useful the method is in practice.

**Post-rebuttal**
The additional experiments including re-training methods are appreciated. However, for me the paper remains a borderline submission. The method works in practice but the analytical analysis/insights are limited.

---

> ### Author Response · Authors · 2022-11-17
> **Response to Reviewer 6kPp**
>
> ## Experimental organization
> Thank you for your constructive comments.
> Following your suggestions, we re-arrange the experiments in Sec. 5. The quantization performance  with only MSE loss is added in Sec. 5.3 (see Fig. 4 and Fig. 5) as a part of the main paper to show the effectiveness of our proposed RDO loss. The original R-D curves in the previous submission are moved to the supplemental materials for your reference, and instead, we use a table (Table 1) in this revision to report the BD-rate loss against the floating-point models.
>
> ## Detailed setup and additional comparison
> It is absolutely insightful for our paper to have more details of the experimental setup and comparisons with those QAT methods (i.e., re-training required). As a result, two typical QAT methods (e.g., [1], [2]) are provided in Fig. 4 and Fig. 5 for illustration. As seen, our method not only needs less runtime with (about 78\% of runtime reduction, e.g., several hours versus tens of hours) but also demonstrates better performance. We enforce the same platform for a fair comparison. More details related to the experimental setup (e.g., used hardware, processing steps) and running time are added in the supplementary materials (see Sec. C) due to the space limitation.
>
> ## How to set calibration dataset
> In our experiment, the calibration dateset is selected randomly from the ImageNet and does not include any test images we used, e.g., Kodak and Tecnick. We also report the performance by changing the size of the calibration set. As illustrated in Fig. 7, using only 10 random images
> that are different from the test set  is sufficient to calibrate our method.
>
> ## The algorithm for every desired bit rate
> Thanks for the comment. This point is not the weakness of our quantization method. We just need an off-the-shelf model for quantization. Given that most popular LICs use separate models for various bitrates, our method is applied accordingly. However, if the method supports variable-rate compression using a single model, our method is applicable as well.
>
> ---
>
> [1] Sun, Heming, et al. "End-to-end learned image compression with fixed point weight quantization." 2020 IEEE International Conference on Image Processing (ICIP). IEEE, 2020.
>
> [2] Sun, Heming, Lu Yu, and Jiro Katto. "Learned Image Compression with Fixed-point Arithmetic." 2021 Picture Coding Symposium (PCS). IEEE, 2021.

---

### Author Response · Authors · 2022-11-26
**Response from Authors**

We thank all the reviewers for their insightful comments. We have tried to response the questions raised, in the response box for respective reviewers. If there are any further questions, please let us know.

---

### Author Response · Authors · 2022-12-07
**Paper2347 Authors Response to Area Chairs and Reviewers**

Dear AC, reviewers:

We sincerely thank the area chairs and reviewers for their time and efforts.

We added extra experiments according to the suggestions of reviewers, among which two reviewers had multiple rounds of interaction with us. However some reviewers did not respond. We hope we can get help to further improve our work.

---

### Decision · Program_Chairs · 2023-01-20

**Decision:**

Reject

**Justification For Why Not Higher Score:**

Too many reviewers whose reviews pointed flaws in the writing or in the experiments did not get swayed by the rebuttal.

**Justification For Why Not Lower Score:**

N/A

**Metareview: Summary, Strengths And Weaknesses:**

This paper studies Post-Training Quantization (PTQ) for Learned Image Compression (LIC) models. The paper shows that minimizing MSE in PTQ is suboptimal, and proposes a novel Rate-Distortion Optimized (RDO) PTQ. The paper lacks some ablations and experimental details to be more useful in future PTQ research work. In its current form, the contribution is rather incremental and would probably better fit a quantization or compression workshop than ICLR.

**Summary Of Ac-Reviewer Meeting:**

N/A